# Deep Metric Learning Network using Proxies for Chromosome Classification in Karyotyping Test

**Author(s) names withheld**                                                                EMAIL(S) WITHHELD

## Abstract

In karyotyping, the classification of chromosomes is a tedious, complicated, and time-consuming process. It requires extremely careful analysis of chromosomes by well-trained cytogeneticists. To assist cytogeneticists in karyotyping, we introduce Proxy-ResNeXt-CBAM which is a metric learning based network using proxies with a convolutional block attention module (CBAM) designed for chromosome classification. RexNeXt-50 is used as a backbone network. To apply metric learning, the fully connected linear layer of the backbone network (ResNeXt-50) is removed and is replaced with CBAM. The similarity between embeddings, which are the outputs of the metric learning network, and proxies are measured for network training.

Proxy-ResNeXt-CBAM is validated on a public chromosome image dataset, and it achieves an accuracy of 95.86%, a precision of 95.87%, a recall of 95.9%, and an F-1 score of 95.79%. Proxy-ResNeXt-CBAM which is the metric learning network using proxies outperforms the baseline networks. In addition, the results of our embedding analysis demonstrate the effectiveness of using proxies in metric learning for optimizing deep convolutional neural networks. As the embedding analysis results show, Proxy-ResNeXt-CBAM obtains a 94.78% Recall@1 in image retrieval, and the embeddings of each chromosome are well clustered according to their similarity.

**Keywords:** Karyotyping test, Karyotype, Chromosome, Metric learning, Proxy, Deep learning

## 1. Introduction

In the field of cytogenetics, karyotyping is one of the most reliable tests for detecting genetic abnormalities (e.g. Down syndrome, Edwards syndrome, Turner syndrome and Chronic myelogenous leukemia). Karyotyping is performed on individual human chromosome images obtained during the metaphase stage of cell division. As shown in Figure 1, a healthy human cell consists of 22 pairs of autosomes and a single pair of sex chromosomes (X and Y), totaling 23 pairs of chromosomes. Cytogeneticists have individually segmented a total of 46 chromosomes of healthy humans, thoroughly examined, and classified them as one of the 24 chromosome types (1, ..., 22, X, Y).

The manual analysis of each and every chromosome for diagnosis purposes takes a considerable amount of time and is highly dependent on expert knowledge. In the recent past years, artificial intelligence researchers have focused on automating the karyotyping process to assist doctors and reduce their work load. Researchers proposed and used various machine learning and deep learning techniques for automating the karyotyping process and obtained

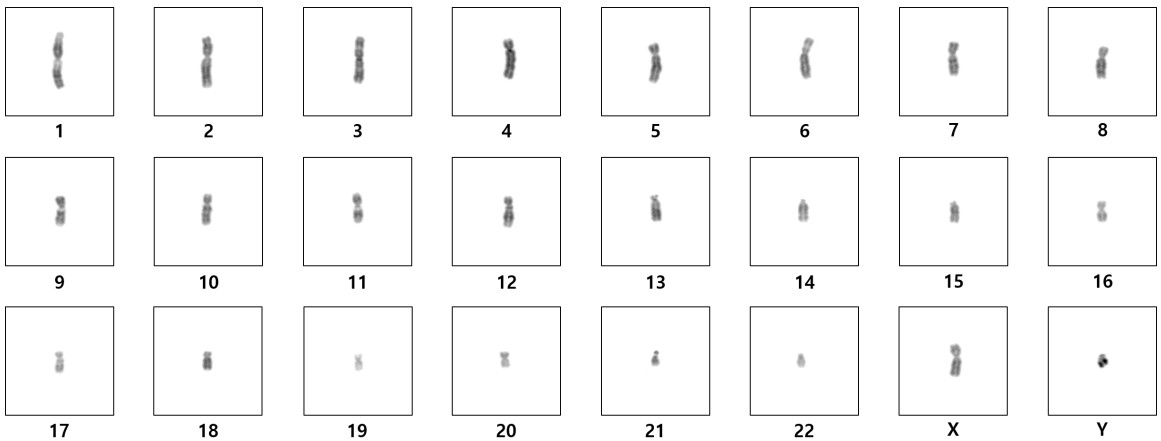

Figure 1: Chromosome images

encouraging results. Most studies used deeply stacked convolutional neural networks for chromosome classification (Hu et al., 2019; Zhang et al., 2018; Sharma et al., 2018b,a, 2017) and some studies employed feature based deep learning methods (Qin et al., 2019; Jindal et al., 2017).

Metric learning is used to convert objects to corresponding embeddings. The main advantage of metric learning is that it can exploit the semantic similarity of objects to regularize a network. In embedding space, objects from the same class are more closer than objects from different classes. In the fields of image retrieval and face verification, the majority of current state-of-the-art (SOTA) approaches are based on metric learning (Omkar M. Parkhi and Zisserman; Zhai et al.). The success of metric learning in these fields is dependent on its ability to understand the similarity of objects.

The main contributions of this paper are as follows. First, we use proxy based metric learning for the chromosome classification task. Second, we introduce a new network for metric learning named Proxy-ResNeXt-CBAM, which has a convolutional block attention module (CBAM). The classification performance of our network is higher than that of conventional deep convolutional neural networks. Finally, we apply image retrieval and image clustering to validate the embedding of chromosomes.

## 2. Methods

**Networks:** ResNeXt (Xie et al., 2017) is simple and highly modularized deep convolutional neural network. ResNeXt consists of a repeating convolutional building block that aggregates feature information. ResNeXt has a multi-branch architecture with only a few hyper-parameters. we use ResNeXt-50 as a backbone network, which is one variation of ResNeXt.

In metric learning, convolutional features of objects are converted into N-dimensional embeddings. Embeddings represent objects from the same class, which are closer in distance than objects from different classes. When modifying a general classification network to

Table 1: Description of networks.

| Network | RexNeXt | Proxy-ResNeXt | Proxy-ResNeXt-CBAM |
|---|---|---|---|
| Backbone | conv1 | | 7×7, 64, stride 2 |
| | | | 3×3, max pool, stride 2 |
| | conv2 | | $\begin{bmatrix} 1\times1,\ 128 \\ 3\times3,\ 128,\ C=32 \\ 1\times1,\ 256 \end{bmatrix} \times 3$ |
| | conv3 | | $\begin{bmatrix} 1\times1,\ 256 \\ 3\times3,\ 256,\ C=32 \\ 1\times1,\ 512 \end{bmatrix} \times 4$ |
| | conv4 | | $\begin{bmatrix} 1\times1,\ 512 \\ 3\times3,\ 512,\ C=32 \\ 1\times1,\ 1024 \end{bmatrix} \times 6$ |
| | conv5 | | $\begin{bmatrix} 1\times1,\ 1024 \\ 3\times3,\ 1024,\ C=32 \\ 1\times1,\ 2048 \end{bmatrix} \times 3$ |
| | | | CBAM |
| | global average pool | global average pool | global average pool + global max pool (concat) |
| | | Layer Normalization | Layer Normalization |
| | 1000-d fc | Normalization | Normalization |
| output size | 1×c | 2048×c | 2048×c |

C = grouped convolutions, c = the number of classes

convert images into embeddings, its fully connected (FC) linear layer, which is the last layer of the network, is removed. Therefore, the output vector size of the network changes from 1×c to N×c (c is the number of classes).

To obtain adaptive embedding vectors, we employ a convolutional block attention module (CBAM)(Woo et al., 2018). CBAM is proved as an effective but simple attention module for deep convolutional neural networks. Since CBAM is a lightweight and general module, it can be seamlessly integrated into metric learning based networks. For our metric learning-based network Proxy-ResNeXt-CBAM, CBAM is attached to the backbone network (ResNeXt-50) which performs feature map extraction. CBAM sequentially infers two separate attention maps. To adaptivly refine attention maps, both attention maps are multiplied to a input feature map.

A proxy (Movshovitz-Attias et al., 2017) is a representative embedding of objects and employed for comparing similarities. We used a cosine similarity based distance metric to calculate losses for optimizing Proxy-ResNeXt-CBAM. Proxies of each class (1, 2, 3, ..., X, Y) are the same size as the embeddings of objects and trained with the network parameters.

Various networks designed for chromosome classification are summarized in Table 1. ResNeXt is the original network. Proxy-ResNext is a metric learning network that employs proxies. Proxy-ResNext-CBAM is Proxy-ResNeXt with the attached CBAM. Cross-entropy loss is used when training ResNeXt, and normalized softmax loss, which is described in the next section, is used to train metric learning based networks. The details of each network are provided in Table 1.

**Normalized Softmax Loss:** When generating embeddings using deep neural networks (DNNs), quantifying the similarity and dissimilarity of objects makes it difficult to optimize networks. Since DNNs use only a mini-batch of objects at each iteration, it is difficult to sample a set of pairs or triplets of objects for optimally generating embeddings. So a set of pairs or triplets of objects has to be sampled from the mini-batch. The object pair sampling method (similar or dissimilar data points) and the object triplet sampling method (Wu et al., 2017; Schroff et al., 2015) are the most commonly used. Contrastive loss (Chopra et al., 2005) and triplet loss (Hoffer and Ailon, 2015) were proposed for the pair and triplet sampling methods, respectively.

A sampling strategy of selecting informative pairs or triplets of objects is necessary for effectively optimizing models and improving convergence rates. We employed the class balanced sampling strategy which is commonly used in image retrieval tasks (Zhai et al.). This strategy involves including multiple objects per class when constructing the training mini batch. For each training batch, classes are selected and objects in each class are chosen.

The normalized softmax loss which is the partially modified conventional loss can be applied in the class balanced sampling strategy. The normalized softmax loss can be used for proxy-based metric learning when the class weight is represented as a proxy and a distance metric can be used as the cosine similarity distance function. We used the same notations as in (Zhai et al.). $x$ denotes the embedding of an input image with the class label $y$. The normalized softmax loss can be expressed with the weight of class $p_y$ among all possible classes in set Z:

$$L_{norm} = -\log\left(\frac{\exp\left(x^T p_y\right)}{\sum_{z \in Z} \exp\left(x^T p_z\right)}\right) \tag{1}$$

## 3. Experiments

### 3.1. Dataset

We utilized the publicly available Bioimage Chromosome Classification dataset (Poletti et al., 2008). This dataset contains a total of 5,256 images of chromosomes of healthy patients, which were manually segmented and labeled by expert cytogeneticists. As done in the baseline methods, we divided the 5,256 images into training (4,176), validation (360), and test (720) sets. In our experiments, the resolution of chromosome images in grayscale is $50 \times 50$, which is enlarged to the desired resolution of $256 \times 256$.

### 3.2. Experiment Settings

**Experimental setups:** In this section, we evaluate the performance of chromosome classification networks using different experimental setups which are summarized in Table 2. We used a Pytorch deep learning framework in our experiments. The output embedding of size N was set to 2048 and output embeddings were compared with the proxies of each class. All the networks were trained using the SGD optimizer with an initial learning rate

of 1e-3, a momentum of 0.9 and an L2 penalty weight-decay of 1e-4. We used a simple learning rate scheduler called ReduceLRonPlateau which is used to reduce the learning rate by a constant factor when the loss on the validation set plateaus. ReduceLRonPlateau was set to a factor of 0.1 for every 10 patients. The best network parameters on the validation set is saved at each epoch and used for testing.

Table 2: Experimental setups

| Experiment Name | Metric Learning with Proxy | Loss Type | CBAM | Backbone Network |
|---|---|---|---|---|
| **ResNeXt** | X | Cross-entropy | X | |
| **Proxy-ResNeXt** | O | Normalized softmax | X | ResNeXt-50 |
| **Proxy-ResNeXt-CBAM** | O | Normalized softmax | O | |

Before feeding an individual chromosome image as input to networks, a) Random Crop, b) Random Horizontal Flip, and c) Random Vertical Flip were applied during the training phase to augment data. Padded images ($256 \times 256$) were randomly cropped to the size of $224 \times 224$. In the testing phase, images were randomly cropped to $224 \times 224$ without any flip augmentation. The TorchVision package was used for this task.

### 3.3. Results

**Chromosome classification performance** We measured and compared the performance of Proxy-ResNeXt-CBAM with that of the baseline classification networks on the Bioimage Chromosome Classification dataset. Accuracy, Precision, Recall, and F-1 score were used as evaluation metrics to measure the classification performance of the networks in our experimental setups. The authors who proposed the baseline classification networks reported only the Accuracy results and not the Precision, Recall or F1 score results in their manuscript; therefore, only the Accuracy results of the baseline networks are listed in Table 3.

For a fair performance comparison, we randomly generated 100 different datasets and measured performance on each dataset (performance of the baseline networks on a single dataset was measured). The average performance of metric learning based networks using proxies and that of the baseline networks are shown in Table 3.

Table 3: Classification performance of various networks

| Method | Accuracy | Precision | Recall | F-1 score |
|---|---|---|---|---|
| Deep CNN (Sharma et al., 2018a,b) | 87.50 | N/A | N/A | N/A |
| ResNet-50 (Sharma et al., 2018a,b) | 87.64 | N/A | N/A | N/A |
| Res-CRANN (Sharma et al., 2018a) | 90.42 | N/A | N/A | N/A |
| Super-Xception (Sharma et al., 2018b) | 92.36 | N/A | N/A | N/A |
| **ResNeXt** | 90.22±4.52 | 89.93±2.68 | 88.91±2.39 | 88.28±2.24 |
| **Proxy-ResNeXt** | 95.30±1.29 | 95.29±1.74 | 95.03±1.05 | 95.02±1.41 |
| **Proxy-ResNeXt-CBAM** | **95.86±0.62** | **95.87±0.61** | **95.90±0.73** | **95.79±0.65** |

We compare the performance of ResNeXt that do not use metric learning, proxies, or CBAM with baseline classification networks. Generally, ResNeXt achieves better performance than conventional ready-made CNNs (e.g. Deep CNN and ResNet-50 (Sharma et al., 2018a,b)), and obtains performance comparable to that of CNNs with modified architectures (e.g. Res-CRANN (Sharma et al., 2018a) and Super-Xception (Sharma et al., 2018b)). However, the high standard deviation values of accuracy, precision, recall and F-1 score of ResNeXt can be attributed to the inconsistent performance of RexNeXt in chromosome classification.

We compared the performance of ResNeXt with that of metric learning networks trained with proxies (Proxy-ResNeXt and Proxy-ResNeXt-CBAM). The chromosome classification performance of Proxy-ResNeXt sharply increased. Proxy-ResNeXt, the simple metric learning based network, achieves an accuracy of 95.3±1.29, a precision of 95.29±1.74, a recall of 95.03±1.05, and an F-1 score of 95.02±1.41. The overall performance of Proxy-ResNeXt is better than that of ResNeXt, and the standard deviations of the accuracy, precision, recall, and F-1 scores of Proxy-ResNeXt are lower than those of ResNeXt. The drastic improvement in performance of Proxy-ResNeXt can be attributed to metric learning. Additionally, the performance of the metric learning based network with CBAM (Proxy-ResNeXt-CBAM) also improved. All the performance metric scores slightly increased as their standard deviations decreased. CBAM which is attached to the last part of Proxy-ResNeXt effectively uses adaptive feature refinement to consistently train the network.

**Embedding analysis** Metric learning is used to train networks to generate consistent object embeddings. During the training phase, the object from the same class are mapped closer to each other than objects from different classes. Therefore, embeddings can be utilized to compute the similarity between objects. The distances between embeddings converted from objects can be compared, which makes it possible to apply metric learning to tasks beyond image classification, such as image retrieval and image clustering. In this section, Proxy-ResNeXt and Proxy-ResNeXt-CBAM are applied in image retrieval and image clustering tasks to determine whether they can effectively generate object embeddings.

Table 4: Image retrieval performance of Proxy-ResNeXt and Proxy-ResNeXt-CBAM

| Method | Recall@1 | Recall@2 | Recall@4 | Recall@8 |
|---|---|---|---|---|
| **Proxy-ResNext** | 94.61±1.35 | 96.22±0.75 | 97.11±0.3 | **97.98±0.08** |
| **Proxy-ResNext-CBAM** | **94.78±0.46** | **96.44±0.4** | **97.36±0.48** | 97.81±0.23 |

Recall@K is used to quantitatively measure the image retrieval performance of Proxy-ResNeXt and Proxy-ResNeXt-CBAM. Recall@K is the ratio of relevant objects found in the top-K retrievals. Each object is converted to a corresponding embedding. Cosine similarity is used to retrieve the top K objects, excluding the query object itself, from the test set. Table 4 shows the image retrieval performance of Proxy-ResNeXt and Proxy-ResNeXt-CBAM in terms of recall at 1, 2, 4 and 8. As demonstrated by their Recall@1 scores, both Proxy-ResNeXt and Proxy-ResNeXt-CBAM are effective in retrieving objects similar to the query object. The Recall@8 score of Proxy-ResNeXt and Proxy-ResNeXt-CBAM increased to about 98%. Proxy-ResNeXt-CBAM obtains better performance than Proxy-ResNeXt, which shows that CBAM can help a network more effectively generate embeddings.

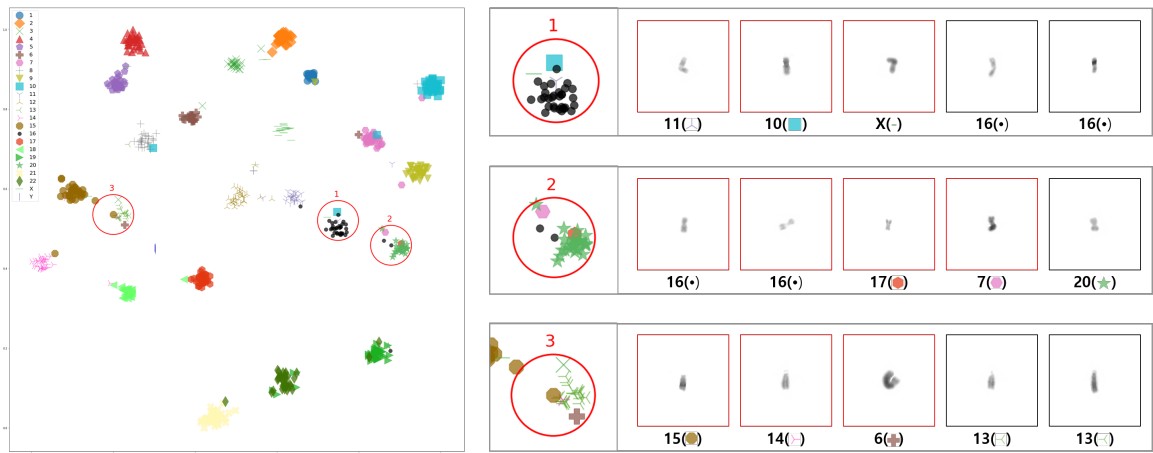

Figure 2: Image clustering

For the qualitative analysis, we clustered embeddings to better visualize embeddings. The t-SNE (t-distributed stochastic neighbor embedding) (Maaten and Hinton, 2008) method which converts similarities of data points into joint probabilities was used to reduce the dimension of embeddings from 2048 to 2. Figure 2 illustrates all the data points that represent embeddings of chromosomes, which are generated by Proxy-ResNeXt-CBAM. Almost all data points from each class are accurately clustered. Three areas that contain some mis-clustered data points are circled in red in the left sub-figure of Figure 2. In the right sub-figure of Figure 2, images of mis-clustered data points are highlighted in red boxes and images of the well-clustered data points are highlighted in black boxes. Most of the images of the mis-clustered data points circled in red are generally in lower resolution than the images of the well-clustered data points. Since chromosomes of different classes have unique band patterns, the band patterns of chromosomes are crucial in classifying chromosomes. But the band patterns of the mis-clustered chromosomes are not clear due to the lower resolution. The lengths of the mis-clustered chromosomes are shorter than that of the well-clustered chromosomes from the original class. In addition, distorted chromosome images degraded the clustring performance of Proxy-ResNeXt-CBAM.

## 4. Conclusion

To assist cytogeneticists with karyotyping and help them more efficiently classify chromosomes, we proposed Proxy-ResNeXt-CBAM which is a metric learning network that has an attached CBAM and uses proxies in chromosome classification. Proxy-ResNeXt-CBAM outperforms conventional classification deep learning networks. Also, we conducted an embedding analysis to demonstrate the effectiveness of using proxies. The embedding analysis results shows that using proxies improves the performance of deep convolutional neural networks in distinguishing embeddings of chromosomes of each class.

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
