# OpenReview forum: "Deep Metric Learning Network using Proxies for Chromosome Classification in Karyotyping Test"
_MIDL.io/2020/Conference — Submitted to MIDL 2020_

### Official Review · AnonReviewer1 · 2020-03-08
**Review of "Deep Metric Learning Network using Proxies for Chromosome Classification in Karyotyping Test"**

**Rating:** 1
**Confidence:** 4

**Summary:**

The paper proposed Proxy-ResNeXt-CBAM, a metric learning network that has an attention mechanism called CBAM and uses proxies in chromosome classification. The goal is to assist cytogeneticists with karyotyping and help them more efficiently classify chromosomes. Their best model outperforms conventional classification deep learning networks.

**Strengths:**

The authors utilized the publicly available Bioimage Chromosome Classification dataset. Results on this benchmark seem promising and outperform some recent baselines. The experimental analysis seems thorough

**Weaknesses:**

- The paper lacks original contributions. Neither deep metric learning with proxy nor CBAM was originally invented. It is thus a typical "existing A + existing B, applied to some new C" type of work.

- The definition of "proxy" is very much unclear from paper. Is that just the hidden features of CNN, optimized under a cosine distance? If so, the authors over-complicated their description and may have over-stated contribution.

- The motivation of CBAM is very unclear: it looks like the authors adopted that only because "the classification performance of our network is higher". Why does it help the proposed metric learning? Why just this specific attention, given the numerous attention mechanisms developed? None of those questions is well justified nor motivated.

- The writeup is not easy to follow, and reading experience is not pleasant. Specifically, the authors seem to often unnecessarily self-repeat, e.g. "We introduce Proxy-ResNeXtCBAM which is a metric learning-based network using proxies ..."" Proxy-ResNeXt-CBAM which is the metric learning network using proxies outperforms ...""Proxy-ResNext is a metric learning network that employs proxies".

**Justification Of Rating:**

See above weakness: 1) lack or original contribution; 2) the definition of "proxy" is very much unclear; 3) the motivation of CBAM is very unclear and not well motivated; 4) the writeup is very sloppy

**Paper Type:**

validation/application paper

**Special Issue:**

no

---

### Official Review · AnonReviewer4 · 2020-03-12
**The justification of the method is unclear**

**Rating:** 2
**Confidence:** 3

**Summary:**

This paper proposes a metric learning based model for chromosome karyotyping. The model learns a proxy embedding for each class, and uses cosine similarity to classify new inputs based on the distance between the input embedding and each proxy embedding.
The authors use a ResNeXt with CBAM to compute image embeddings, which are then compared to the class proxies.

**Strengths:**


+ The paper achieves state of the art results on a public chromosome classification dataset.
+ Improves previous results by a significat margin.
+ The authors show an analysis of the image embeddings computed by their model.

**Weaknesses:**


My main concern with this paper is that most of the modelling decisions are barely justified.
1. The authors use CBAM "to obtain adaptive embedding vectors". What exactly does that mean? Adaptive to what? Why does it help the model to solve the task?
2. It is unclear why the authors approach the chromosome classification task as a metric learning problem. The paper says that "The main advantage of metric learning is that it can exploit the semantic similarity of objects to regularize a network" but that is not explained or backed with experiments.
3. As mention in the paper, face verification and image retrieval are successful applications of metric learning. However, these tasks can't be approached as a classification problem, as opposed to chromosome classification. What is the reason for a metric learning approach to be better than a classification model if the task can easily be framed as a classification problem?
4. Learning the proxies and applying softmax on the dot product between each proxy and each sample embedding is like learning a classifier on the embeddings, where the weights of the classifier are given by the proxies. Then, does the improvement come from the sampling strategy mentioned in the paper?

**Justification Of Rating:**

Even though the results are good, the experimental section is a bit poor, and should be improved to show where does the improve in performance come from. It is not clear why the proposed model is better than previous approaches, as it is not compared with a standard model that learns a classifier on top of the CBAM embeddings.

**Paper Type:**

methodological development

**Questions To Address In The Rebuttal:**

* What is the justification of using CBAM in this model?
* Why approach the task with metric learning instead of approaching it with a classifier model?

**Special Issue:**

no

---

> ### Author Response · Authors · 2020-03-26
> **Answer**
>
> Thank you for your effort to read our paper and helpful comments.
>
> Here is our answers of your comments.
>
> 1) What is the justification of using CBAM in this model?
>
> ::Deep metric learning aims to measure the similarity between images based on their convolutional embeddings, and uses an optimal distance metric for learning tasks. Embeddings of images from the same class are closer in distance than embeddings of images from different classes. Therefore, accurately converting images into embeddings is crucial. Therefore we employ CBAM which is proved to generate attention maps which is effective to represent images in deep convolutional neural networks.
>
>
> 2) Why approach the task with metric learning instead of approaching it with a classifier model?
>
> ::In karyotyping task, inter-class type karyograms are highly similar but slightly different. In addition, the ‘resolution’ of chromosomes which means the length of karyograms makes karyotyping difficult. In the cytogenetics field, long chromosomes with band patterns that cytogeneticists can clearly identify are called 'high resolution chromosomes,' but short chromosomes with compressed band patterns are called 'low resolution chromosomes.' The 'resolution' of chromosomes can vary depending on the stage of cell division, even if the chromosomes are collected from a single patient.
>
> Generally metric learning outperforms in the tasks which is needed to distinguish objects with similar shape but different. Therefore, metric learning is widely used in face recognition tasks and content based image retrieval tasks. Since we thought karyotyping task is close with these type of tasks, we approach it with the metric learning.
>
>
>
>
>
> Additionally, here is our opinion to the weaknesses which are not handled in 'Questions To Address In The Rebuttal' .
> 1- The authors use CBAM "to obtain adaptive embedding vectors". What exactly does that mean? Adaptive to what? Why does it help the model to solve the task?
>
> :: As you know, CBAM consists of the channel-wise module and the spatial wise module. In each module, CBAM has two pooling layers, the global max pooling and the global average pooling. The max pooling obtains most remarkable one feature and the average pooling obtains comprehensive one feature. Therefore, using both features is better to represent chromosomes than using features only from global average pooling.
>
>
> 4- Learning the proxies and applying softmax on the dot product between each proxy and each sample embedding is like learning a classifier on the embeddings, where the weights of the classifier are given by the proxies. Then, does the improvement come from the sampling strategy mentioned in the paper?
> :: Sampling strategy helps to learn the model efficiently by sampling the classes balanced. therefore, it can do fast convergence during the training.
>
>
>
> We wish our answers and opinions can help you understand more precisely.
>
> Thank you.

---

### Official Review · AnonReviewer2 · 2020-03-13

**Rating:** 2
**Confidence:** 4

**Summary:**

the authors use a metric learning approach for  chromosome classification task. They use the Proxy Ranking loss to train the embedding function. They augment the embedding function with a convolutional block attention module (CBAM), which uses a combination of channel attention as well as spatial attention modules.

They compare their method on other state-of-the-art methods on that dataset and perform some ablation study.

**Strengths:**

The authors compare their method with other deep learning methods and also perform some ablation study especially on different components of the model.

Table 2 explaining the experimental setup is quite helpful.


**Weaknesses:**

1- The motivation of the paper: In the abstract, it is noted: "In addition, the results of our embedding analysis
demonstrate the effectiveness of using proxies in metric learning for optimizing deep convolutional neural networks."
This puts the emphasis on the proxy-based metric learning approaches. However, there is no ablation study or comparison with other metric learning approaches.
2- There is no good motivation for why CBAM layer is used and what it does. The only explanation is ". CBAM sequentially infers
two separate attention maps. To adaptively refine attention maps, both attention maps are multiplied to a input feature map." I assume the authors are referring to the channel-wise and spatial wise attention modules.  Even so, an intuition behind using CBAM in a metric learning embedding function is unclear.
3 - Better explanation of objective functions and the whole training procedure is needed.



**Justification Of Rating:**

While I think this is an interesting paper, I dont think it has enough contribution to be accepted at MIDL.  I think its more suited for a workshop.
I would recommend the authors to provide more intuition or justification for why the attention module is useful.

**Paper Type:**

validation/application paper

**Questions To Address In The Rebuttal:**

How is the class weights p_y set?

When doing the ablation study of using and not using CBAM do the two models have the same number of parameters?




**Special Issue:**

no

---

> ### Author Response · Authors · 2020-03-26
> **Answer**
>
> Thank you for your effort to read our paper and helpful comments.
>
> Here is our answers of your comments.
>
> 1) How is the class weights p_y set?
>
> ::The class weights (p_y) are the proxies. They represent resulting subspaces of class embeddings. In this study, there is a total of 24 proxies for classes (1, 2, 3, ..., X, Y). And each of them has 2048 dimension which is same as the output size of networks.
>
> 2) When doing the ablation study of using and not using CBAM do the two models have the same number of parameters?
>
> ::As you know, CBAM sequentially infers two separate attention maps inside the network. One is made by the global max pooling and the other is made by the global average pooling. And both of them are concatenated. Therefore, Using CBAM has little bit more parameters.
>
>
>
>
> Additionally, here is our opinion to the weaknesses.
> 1- The motivation of the paper: In the abstract, it is noted: "In addition, the results of our embedding analysis demonstrate the effectiveness of using proxies in metric learning for optimizing deep convolutional neural networks."
> This puts the emphasis on the proxy-based metric learning approaches. However, there is no ablation study or comparison with other metric learning approaches.
>
> :: This is the first study that metric learning is applied on the public dataset. All the baseline studies used basic convolutional neural networks. Therefore we compared the classification performance with them.
> In addition, we reported ablation study results of our method. Classification performance of networks which use metric learning method and which does not use metric learning method are listed in Table 3.
>
> 2- There is no good motivation for why CBAM layer is used and what it does. The only explanation is ". CBAM sequentially infers two separate attention maps. To adaptively refine attention maps, both attention maps are multiplied to a input feature map." I assume the authors are referring to the channel-wise and spatial wise attention modules.  Even so, an intuition behind using CBAM in a metric learning embedding function is unclear.
>
> :: As you know, CBAM consists of the channel-wise module and the spatial wise module. In each module, CBAM has two pooling layers, the global max pooling and the global average pooling. The max pooling obtains most remarkable one feature and the average pooling obtains comprehensive one feature. Therefore, using both features is better to represent chromosomes than using features only from global average pooling.
>
> 3 - Better explanation of objective functions and the whole training procedure is needed.
> :: As we explain the proxies in the above answer, proxies has same dimension as the output of network(embbedings). Since we used similarity based soft max loss, one embedding and 24 proxies are compared based on the cosine similarity in the training phase.
>
>
> We wish our answers and opinions can help you understand more precisely.
>
> Thank you.

---

### Meta-Review · Area_Chair1 · 2020-04-04
**MetaReview of Paper331 by AreaChair1**

**Rating:** 2

**Metareview:**

All three reviewers seem to agree on the fact that the results presented by the paper  are promising and outperform the results from recently published baselines on a public chromosome classification dataset.  However, the main issue of the paper is a lack of novelty since main contributions (deep metric learning with proxy and CBAM ) have been proposed before.  In addition, the paper shows little motivation for deep metric learning with proxy and CBAM.  The rebuttal tries to address the motivation issues mentioned above, but not very successfully given that there is a very pragmatic explanation without any rational decision process that can justify the use of deep metric learning with proxy and CBAM.  One reviewer also mentions that the paper does not explain the objective functions and the whole training procedure.  Given these issues and a rebuttal that did not answered the reviewers' questions, I do not recommend this paper for acceptance.

**Paper Type:**

validation/application paper

**Special Issue:**

no

---

### Decision · Program_Chairs · 2020-04-11

Reject